# Ampliseq for Illumina Technology Enables Detailed Molecular Epidemiology of Rabies Lyssaviruses from Infected Formalin-Fixed Paraffin-Embedded Tissues

**DOI:** 10.3390/v14102241

**Published:** 2022-10-12

**Authors:** Susan Angela Nadin-Davis, Allison Hartke, Mingsong Kang

**Affiliations:** Centre of Expertise for Rabies, Ottawa Laboratory Fallowfield, Canadian Food Inspection Agency, Ottawa, ON K2H 8P9, Canada

**Keywords:** rabies lyssavirus, viral typing, phylogeny, formalin fixed tissues, viral lineage

## Abstract

Whole genome sequencing of rabies lyssaviruses (RABVs) has enabled the generation of highly detailed phylogenies that reveal viral transmission patterns of disease in reservoir species. Such information is highly important for informing best practices with respect to wildlife rabies control. However, specimens available only as formalin fixed paraffin embedded (FFPE) samples have been recalcitrant to such analyses. Due to the damage inflicted by tissue processing, only relatively short amplicons can be generated by standard RT-PCR methods, making the generation of full-length genome sequences very tedious. While highly parallel shotgun sequencing of total RNA can potentially overcome these challenges, the low percentage of reads representative of the virus may be limiting. Ampliseq technology enables massively multiplex amplification of nucleic acids to produce large numbers of short PCR products. Such a strategy has been applied to the sequencing of entire viral genomes but its use for rabies virus analysis has not been reported previously. This study describes the generation of an Ampliseq for Illumina primer panel, which was designed based on the global sequence diversity of rabies viruses, and which enables efficient viral genome amplification and sequencing of rabies-positive FFPE samples. The subsequent use of such data for detailed phylogenetic analysis of the virus is demonstrated.

## 1. Introduction

Rabies diagnosis is normally performed by detection of the rabies virus antigen in fresh brain tissue using immunofluorescence. This technique, referred to as the direct fluorescent antibody (DFA) test, is in wide application due to the availability of specific antibodies and published protocols [1]. Additional information about the nature of the infecting strain can also be derived by indirect immunofluorescent procedures which use a panel of monoclonal antibodies exhibiting selective binding to various viral strains [2,3]. Viral typing can be of significant value for attribution of the infection source in areas harboring syntenic reservoir hosts or regions previously free of the disease as this information informs effective rabies control strategies. Alternatively, the detection of viral RNA using RT-PCR methods, particularly those based on real-time formats, have gained considerable acceptance as sensitive and robust primary diagnostic tools [4,5], while the sequencing of longer amplicons generated through standard RT-PCR methods readily provide viral typing information [6]. Indeed, molecular epidemiological analysis has enabled global rabies lyssavirus (RABV) classification into seven major lineages: Cosmopolitan, Asian, Africa-2 and Africa-3, Arctic and Arctic-related, Indian Subcontinent, and the most divergent American Indigenous. Some of these lineages are further subdivided into multiple variants, each of which circulates in a specific host and geographical area [7]. Such studies provide significant insights into viral evolution and spread, especially with the use of highly parallel sequencing technologies to characterize whole viral genomes [8,9].

However, tissue is occasionally formalin fixed prior to the suspicion of rabies virus infection and when a diagnosis of such a specimen is required alternative methods are necessary. After paraffin embedding and sectioning of such samples, viral antigen can be detected by immunohistochemical techniques [10,11] although these often do not consistently provide similar sensitivity to the gold standard immunofluorescent protocol. Moreover, historically these samples have not been readily amenable to viral typing. In situ hybridization methods, which evolved from immunohistochemical procedures, were developed as useful diagnostic tools for the detection of viral RNA [12] and this testing format even provided a means of viral strain typing when discriminatory probes were employed [13]. However, like the immunohistochemical methods themselves, these tools were labor intensive and available in only a few laboratories. The development of PCR technology spurred efforts to apply this technique for rabies diagnosis on FFPE tissues and successful diagnosis was found to depend on both careful RNA extraction and limits on the length of the targeted amplicon due to the damage inflicted on the RNA target by the fixation process [14]. These limitations have now largely been addressed with the development of real-time RT-PCR (RT-qPCR) methods that amplify short sequences and can accurately diagnose the disease as well as yield limited sequence data to provide a probable viral type [15]. However, the full characterization of rabies virus genomes using such samples remains challenging. Amplicon length limitations make the generation and sequencing of multiple individual PCR products highly tedious. The use of whole tissue shotgun sequencing by next generation sequencing methods has been reported for fresh tissue [16]. While such an approach targets short stretches of RNA and might thus be suitable for FFPE tissue, the relatively high proportion of the host sequence thus generated could severely limit viral genome coverage, especially when starting with relatively small amounts of material.

The concept of Sanger sequencing of multiple PCR products together in a single reaction was first reported in 2002 [17]. Further development of this approach, now known as Ampliseq, employs a panel of PCR primers to amplify multiple specific targets in a single multiplex reaction followed by sequencing of all products as a single sample, as reported for analysis of HIV-1 drug resistance [18]. With careful primer design to ensure specific target amplification, Ampliseq protocols that support use of the two most common highly parallel sequencing platforms, Ion Torrent and Illumina, have subsequently been developed for a variety of applications including complete genome characterization of the RNA virus SARS-CoV-2 [19]. As the amplicons generated by this approach tend to be short (<400 bp), the Ampliseq strategy is highly appropriate for the characterization of nucleic acids recovered from FFPE samples; indeed, targeted transcriptome analysis of FFPE tissues has been achieved [20] and there are several reports applying Ampliseq technology to FFPE tissue samples for subtyping of several different types of human cancers [21,22].

This report describes a protocol to generate extensive RABV sequence information using a MiSeq instrument from archived FFPE tissues infected with a variety of RABV variants. The methodology involves processing of FFPE tissue to extract RNA suitable for robust viral RNA detection by RT-qPCR, and the application of a universal RABV Ampliseq for Illumina panel to generate amplicons covering a significant portion of the genome for parallel sequencing. Use of these sequence data to infer viral type and perform detailed epidemiological analysis is demonstrated. This methodology can unlock valuable information contained in archival material relevant to the emergence and evolution of this important pathogen.

## 2. Materials and Methods

### 2.1. Rabies-Positive Unfixed Samples

From a collection of rabies-positive brain tissues, compiled at the Centre of Expertise for Rabies over many years, a cohort of 53 unfixed samples representing the seven RABV lineages that circulate world-wide was selected for proof of principle evaluation of the Ampliseq for Illumina panel. All these samples had been confirmed as rabies positive by DFA test and characterized by genetic methods of viral typing, details of which are summarized in Appendix A.

### 2.2. Rabies-Positive FFPE Samples

A total of 23 rabies-positive submissions to the Centre of Expertise for Rabies which had been characterized antigenically and genetically using established inhouse methods [2,23,24,25] were selected for this study based on their representation of all viral types circulating in Canada (Appendix A). Each isolate was passaged in mice according to standard methods [26]. Animals were euthanized upon presentation of clinical signs and the brains fixed in formalin for 1–2 days prior to paraffin embedding. To prepare FFPE samples for RNA extraction, between three and six 10 µm sections of each block were cut using a microtome and placed in a 1.5 mL microfuge tube. Between each sample the microtome was treated with the cleaning agent Histo-Clear II (Diamed Lab Supplies Inc., Mississauga, ON, Canada) followed by 70% ethanol and the blade was changed. Seven submissions were processed in either duplicate or triplicate to provide multiple samples for determining the repeatability of the procedure thereby yielding a total of 33 separate sequences.

### 2.3. RNA Extraction

Total RNA was recovered from unfixed rabies-infected tissue using TRIzol (Thermo Fisher Scientific, Burlington, ON, Canada) as per the manufacturer’s instructions. The final pellet was dissolved in sterile water, RNA concentration was determined using a NanoVue instrument (GE Healthcare, Chicago, IL, USA), and samples were stored at −80 °C.

Each FFPE sample tube received the following reagents, available from QIAGEN (Toronto, ON, Canada), in order: 200 µL ATL buffer, 20 µL proteinase K, and 160 µL deparaffinization solution. Immersion of the sample in the liquid was ensured using a disposable plunger as needed. Tubes were briefly vortexed and then incubated at 60 °C for 45 min followed by 80 °C for 30 min with agitation at 300 rpm in an Eppendorf Thermomixer C unit (Thermo Fisher Scientific). Samples were allowed to cool and 150 µL of the bottom phase was retrieved for RNA extraction, performed using a MagMax-96 instrument (Thermo Fisher Scientific) and an Ambion AM1830 total RNA extraction kit (Thermo Fisher Scientific) as detailed by the manufacturer. The final RNA sample was recovered in 50 µL elution buffer and RNA concentration was determined spectrophotometrically using a NanoVue instrument. Samples were brought to 6 ng/µL final concentration prior to storage at −80 °C.

### 2.4. RT-qPCR

To establish recovery of amplifiable RNA, samples were tested using a rabies-specific RT-qPCR performed essentially as described previously [27] employing as a template either 0.1 µg total brain RNA from unfixed samples or 8 µL RNA extract (48 ng total RNA) of FFPE samples. For all but three samples, for which inadequate sample was available, the RT-qPCR confirmed a positive result with Ct values ranging between 7 and 28 (unfixed samples) and 13 to 20 (FFPE samples).

### 2.5. Design of Ampliseq for Illumina Panel

A collection of 190 whole genome RABV sequences were recovered from GenBank (Appendix A). While this collection covered the global diversity of the virus, it was heavily represented by the variants native to the Americas since these were our initial primary focus. These sequences were submitted for design of an Ampliseq for Illumina custom RNA primer panel using Illumina inhouse protocols. This involved the generation of a consensus genome sequence (Appendix A) and the use of the negative strand of this sequence to design an optimal single pool of 47 primer pairs. All primers were 25 bases in length and were predicted to generate a range of amplicon sizes of 152–377 bp. A schematic of the consensus sequence illustrating the genomic regions contained within the internal sequence of the 47 amplicons is shown in Figure 1 and location details of this primer panel are provided in Appendix A. An overlap of all contiguous primer pairs was predicted to generate overlaps of internal sequences ranging from 6 to 46 bases in length with an overlap of 11 bases for the majority of amplicons. Successful generation of all amplicons of this panel from a viral template would provide a genome coverage of 98.16% due to lack of information at the two termini. The ordering of this panel from Illumina can be arranged by using the following information: design IDL 160200, sol ID: IAAQ177123_200.

### 2.6. Ampliseq for Illumina Protocol

RNA extracted from unfixed tissue was diluted first to 10 ng/µL with confirmation of the concentration using a Qubit 2 fluorimeter (Thermo Fisher Scientific) with a Qubit RNA broad range assay kit (Thermo Fisher Scientific). The concentration of the RNA recovered from FFPE samples (6 ng/µL) was also verified using the Qubit 2 fluorimeter. Finally, samples were further diluted to 2 ng/µL just prior to use. Five µL of each sample (10 ng) was then processed using an Ampliseq Library Plus for Illumina kit (Illumina, San Diego, CA, USA) together with the custom RNA panel according to chapter three of the manual “Ampliseq for Illumina custom and community panels” (available at https://support.illumina.com/sequencing/sequencing_kits/ampliseq-library-plus-for-illumina/documentation.html, accessed on 12 March 2019). In brief, following a reverse transcriptase step the virus was amplified prior to partial amplicon digestion and indexing for a single primer pool using Ampliseq CD indexes set A (Illumina). The library was then purified using AMPure XP beads (Beckman Coulter, Mississauga, Ontario, Canada), amplified and recleaned prior to evaluation. Sample aliquots were used to determine (1) the DNA concentration (ng/µL) using a Qubit 2 instrument with a dsDNA HS assay (Thermo Fisher Scientific), and (2) the amplicon profile was analyzed on a QIAxcel instrument (QIAGEN) operated as per the manufacturer’s directions using a QIAxcel DNA fast analysis kit (QIAGEN). Based on these analyses, the sample’s nanomolar concentration was estimated and samples were normalized by dilution to a final concentration of 10 nM. Samples with a concentration below this value were used without further dilution. Finally, libraries, each comprised of 48 pooled samples, were run on a MiSeq sequencer (Illumina) using a 2 × 250 bp MiSeq Reagent Kit v2 (Illumina).

### 2.7. Species Composition of Illumina Reads

The proportion of Illumina reads representing the targeted RABV and the host species of origin was explored for seven selected samples. The raw reads were analyzed using Kraken2 classifier v2.1.2 [28] based on a custom database containing RefSeq complete viral genomes and proteins, 2319 genome sequences of rabies virus (Appendix A), and genomes of the following hosts: *Vulpes lagopus* (GCF_018345385.1), *Eptesicus fuscus* (GCF_000308155.1), *Bos taurus* (GCF_002263795.1), *Vulpes* (GCF_003160815.1), *Spilogale gracilis* (GCA_004023965.1), as a proxy for *Mephitis*, and *Canis lupus* (GCF_014441545.1). The results of the taxonomic classification of the raw reads were summarized and visualized using R package Pavian v1.0 [29] prior to importation and presentation in Microsoft Excel365.

### 2.8. Reference-Guided Assembly and Viral Type Determinations

Genomic sequences of test samples were reconstructed from Ampliseq for Illumina reads using reference-guided assembly. For each sample, an initial analysis involved use of the RABV consensus sequence employed for panel design (Appendix A) as the reference for a reference guided assembly (assembly A1). Briefly, sequencing reads weremapped to the reference using minimap2 (version 2.23) [30] with default parameters and SAMtools, version 1.9, (available at http://www.htslib.org) was used to convert the M files to the sorted BAM files. UGENE, version 40.1, (available at https://bio.tools/ugene) was then used to inspect the mapping files and generate the assembly. The final assembled genome was recovered in FASTA format in a single contiguous sequence including gaps for those base positions with zero coverage relative to the reference sequence. An alignment of all sample assemblies with the consensus sequence was generated in MEGA, version X, software (available at https://www.megasoftware.net/older_versions, accessed on 23 August 2019) (Appendix A).

Given the observed variation in genome coverage between samples, a second assembly (A2) was performed on the complete dataset using a sequence generated from a Canadian bat RABV variant (accession #JQ685920), a member of the American Indigenous lineage as reference. This assembly confirmed the significant impact of the reference sequence employed for read assembly on genome coverage (Appendix A). To better evaluate the most appropriate reference sequence for assembly, a region of the genome that yielded good coverage for most samples was identified from the A1 assembly alignment. Many samples generated significant sequence across a portion of the L gene corresponding to base positions 7396–8603 of the consensus sequence and this 1208 base region of the genome was targeted for BLAST analysis. For some samples lacking parts of this sequence a smaller sequence around this region was employed (Appendix A). A standard nucleotide BLAST was undertaken to identify the closest match in GenBank’s lyssavirus rabies collection and thereby infer a likely viral type. Guided by this information, additional reference-guided assemblies were performed on 28 samples using appropriate reference sequences as informed from the BLAST analysis, summarized as follows: Six samples from terrestrial hosts infected by the American Indigenous lineage were reassembled using a mid-Atlantic raccoon RABV sequence, accession #EU311738.3, (assembly A3); eleven bat samples infected by the American Indigenous lineage were reassembled using a *Lasiurus* RABV variant, accession #JQ685902.1, (assembly A4) while another three samples apparently of the vampire bat or free-tailed bat RABV types were reassembled using sequence of a vampire bat variant, accession #AB519642.1, (assembly A5); three Asian samples were reanalyzed using a RABV sequence from the Philippines, accession #EU293111.1, (assembly A6); two Africa-3 samples were reassessed using an Africa-3 RABV sequence, accession #MG458308.1, (assembly A7) and the three samples from Sri Lanka were reassembled using a RABV variant from this nation, accession #AB635373.1 (assembly A8). An additional reference guided assembly was also later undertaken on a sample subset representing those of the Arctic lineage using a Canadian Arctic RABV sample NT.1993.0669AFX, accession #MN233954, as reference (assembly A9).

### 2.9. Phylogenetic Analysis

Phylogenetic analyses were completed using MEGA version X software. Using the reference-guided assembly of each sample that resulted in optimal genome coverage, an alignment of all samples together with another 100 representative RABV genomes and an Australian bat lyssavirus (ABLV) genome as outlier recovered from the NCBI database (Appendix A) was compiled. This positive sense alignment was manually reviewed with deletion of genomic termini that were poorly covered by the test samples, including the first 22 bases and a section of the 3′ terminus, resulting in a final alignment of 11,777 bases. This dataset was employed to generate a neighbor-joining (NJ) tree using 1000 bootstrap replicates and pairwise deletion of gaps and ambiguous bases. A second phylogeny was similarly generated from the optimized Arctic dataset (13 samples) together with 39 reference sequences described previously and an Arctic-related sample (99001NEP) as outlier.

### 2.10. Statistical Analysis

Graphpad Prism 7 (available at https://www.graphpad.com) was used for all statistical analysis. Differences in genome coverage between unfixed samples and FFPE samples were analyzed using the Mann–Whitney U test, while differences in genome coverage of unfixed or FFPE samples using different references for assemblies were calculated by the Wilcoxon matched-pairs signed rank test. A *p*-value < 0.05 was considered statistically significant.

## 3. Results

### 3.1. Amplicon Size Distribution Analysis

#### 3.1.1. Unfixed Samples

The amplicon profiles generated by the Ampliseq for Illumina protocol for selected unfixed samples are shown in Figure 2. As summarized in Appendix A, the panel was expected to generate a series of amplicons in the 200–400 bp range. However, it is clear from the profiles in Figure 2 that, while many of the products fell within this range, products larger than 400 bp were generated for several samples. In some cases, including the two Asian RABV samples (lanes 11 and 12) and the American Indigenous RABV samples (lanes 19 and 20), the products were predominantly of longer size. This raised the question of whether mismatch of some primers of the panel failed to support amplification (due to differences in primer affinities across different variants and lineages) so that only primer pairs that result in longer fragments efficiently amplify the viral target sequence. Alternatively, although the Ampliseq for Illumina panel was expected to be reasonably specific for RABVs, these amplicons could result from amplification of the host sequence. The profiles generated by the samples infected with genetically related viruses were often quite similar (cf. lanes 1, 2, and 4 representing the Cosmopolitan lineage and lanes 17 and 18 representing variants A2 and A4 of the Arctic lineage). However, this did not always hold true; lane 16 representing the more divergent variant Arctic1 of the Arctic lineage gave a distinct profile. As the host species for this Arctic1 variant sample was a skunk while the other Arctic RABVs were recovered from fox species, it was unknown to what extent these profiles were influenced by variation in either the RABV sequence or the host genome. It was notable that samples that generated a Ct value > 20 in the RT-qPCR yielded much fainter amplicon profiles. This suggests that when employing samples with a lower viral load, use of higher quantities of total RNA extract than that recommended in the Ampliseq for Illumina protocol might be beneficial. Furthermore, regardless of Ct value, many samples representative of American bat RABV variants, especially those of *Myotis* hosts, generated weak or unobservable amplicon profiles, suggesting that the panel was suboptimal for these viral types.

#### 3.1.2. Comparison of Unfixed and FFPE Samples

Next, the performance of the Ampliseq for Illumina panel was assessed using the FFPE samples. As it was of particular interest to compare results using unfixed and FFPE samples representative of the same viral variant, the outcome for six of the RABV variants commonly encountered in Canada is shown in Figure 3. Unfortunately, the same initial physical sample could not be used for both analyses. To try to control for the difference in tissue state, samples within 3.5 Ct values, as determined by RT-qPCR, were paired wherever possible. However, in the case of the *Lasiurus* bat variant the only available samples had Ct values differing by 8.5 but the higher Ct of the FFPE sample was still well below the value of 20 and amplified well. In general, the profiles for each of these variants was similar for both tissue types although the FFPE profile was often less intense, especially for samples producing larger amplification products. Indeed, the RNA fragmentation that occurs during the fixation process will tend to preclude the generation of the larger amplicons in this sample type compared to the unfixed samples.

### 3.2. Species Assignment of Illumina Reads

To address some of the questions raised above regarding the specificity of the Ampliseq for Illumina panel, the species assignments of the raw reads for seven of the unfixed tissue samples were analyzed. These samples originated from hosts typical of many of the specimens analyzed and included three Arctic RABVs recovered from a skunk, a red fox, and an arctic fox, two Asian RABV samples, both recovered from dogs, and two American Indigenous RABVs recovered from a big brown bat and a bovine. A summary of the Kraken 2 analysis (Table 1) clearly indicates that >97% of reads from all samples represented RABV sequences and that host sequences made up <2% of all reads; indeed, in some cases the host sequence was barely detectable. These results suggest that the amplicon patterns generated by the panel are not significantly impacted by the host of origin. Longer than expected amplicons appear to be primarily due to the pairing of primers more distant than contiguous pairs.

### 3.3. Amplicon Sequence Analysis

Genome coverage varied widely when using the consensus RABV genome sequence (Appendix A) as the reference for a reference-guided assembly for all samples (Appendix A). More specifically, for the unfixed samples, genome coverage ranged from 39% for an isolate from Sri Lanka to 98% for an isolate in the Africa-1 group while for the FFPE samples the range varied from 26% for a *Lasiurus* bat sample to 95% for an isolate of the NCSK/WSK variant (Figure 4 and Appendix A). The Mann–Whitney U test demonstrated a significant difference between the genome coverage in the two sample types (*p* < 0.0001), which we believed was due to biases introduced by the reference sequence selected, given that the FFPE sample set was extensively represented by the American Indigenous (AI) lineage. As expected, using a reference sequence from a member of the AI lineage (accession #JQ685920) for a reference-guided assembly generated significantly different genome assemblies, again with a very wide range of genome coverage from 15% for an FFPE sample of the Arctic lineage to 93% for two unfixed samples of the AI lineage (Figure 4 and Appendix A). In general samples belonging to the *Eptesicus fuscus* (EF) and *Myotis* (MYO) RABV variants exhibited significantly improved genome coverage while most other samples showed either little improvement or greatly reduced coverage. It was therefore apparent that the choice of reference sequence had a major impact on the extent of genome coverage that could be extracted from the sequence reads and a strategy to improve sequence recovery was clearly needed.

An alignment of the sequences of all test samples generated from assembly A1 (Appendix A) revealed that while all samples exhibited little if any coverage through either the highly divergent GL intergenic region or the 3′ coding terminus of the L gene, more conserved regions of the L gene quite consistently yielded some sequence coverage, in particular the region corresponding to bases 7396–8603 of the consensus sequence. Sequence from this region was employed for a BLAST analysis of each test sample so as to identify a probable viral type and thereby identify a suitable RABV sequence for use in further reference-guided assemblies. The results of these BLAST searches (Appendix A) enabled accurate identification of the viral lineage for 80 samples. Of the six samples that gave an incorrect lineage, five were from the FFPE material of *Lasiurus* and silver-haired bat types that had yielded low coverage in this assembly and yielded only short sequences (>500 bases) for the BLAST analysis while the remaining sample (V854) was a Mexican skunk sample prepared from nonfixed tissue. BLAST reassessment of these samples using the sequences generated from the A2 assembly correctly identified all the bat-associated samples as belonging to the American Indigenous lineage. Of the other 80 samples, the BLAST analysis accurately predicted the RABV variant of 71 of them. Identity values <90% or identities based on shorter sequences (<500 bases) were most prone to generate best matches that were inaccurate in terms of the viral type. Thus, while the predictive ability of this approach could not be considered highly accurate, this information identified appropriate reference sequences for additional assemblies that would optimize the sequence recovery. Accordingly, 28 samples were subject to additional reference-guided assembly as detailed in the Methods section.

### 3.4. Phylogenetic Analysis

To gauge the utility of these assembled sequence data to accurately identify the lineage and viral type of the test samples, an alignment composed of the optimized assemblies for all 86 test samples together with 100 additional reference RABV sequences and an Australian bat lyssavirus (ABLV) sequence as outlier was generated (Appendix A) and employed for phylogenetic analysis (Figure 5). Figure 6, Figure 7 and Figure 8 provide detailed illustrations of specific clades within this tree.

These trees illustrate that the assembled Ampliseq reads clustered, with just one exception, within the expected lineage. The Mexican skunk sample (V854) was anomalous in that the previous N gene analysis had placed it in the CMSK variant but in this analysis, it was clustered within the Cosmopolitan lineage, typical of the South Baja California skunk (SBCSK) variant of Mexico. It remains unclear if this was a sampling error during processing or a prior misidentification. Interestingly, in this phylogeny sample V647 clustered with strong support (bootstrap value of 100%) with this Mexican isolate. Sample V647, which originated from a cougar in California, had remained refractory to genetic typing using primers successful with the expected California skunk (CASK) viral subtype [31] despite a positive DFA result. This close genetic similarity of these two samples clearly suggests the circulation of a Mexican skunk variant in California. While this observation may not be surprising given the geographic proximity of the two locations, viral typing had not previously identified the cocirculation of these two variants in the USA. Unfortunately, there were no complete Puerto Rican samples for inclusion in this phylogeny, but the two Puerto Rican samples of this study (V508 and V522) clustered within the Cosmopolitan lineage, similar to samples that had been partially sequenced previously [32]. Notably the mongoose samples from three separate Caribbean islands (Puerto Rico, Cuba (V1061), and Grenada (GREN RV2854)) identified as separate variants within the Cosmopolitan lineage consistent with reports of independent rabies introduction into these countries. In addition, the consistency of these lineage groupings was further demonstrated by the Nigerian dog sample (V463) which clustered within the Vaccine 2 clade as reported previously [33].

Efforts had been made to reach a genome coverage of 60% for all samples and while many samples exceeded this value significantly, seven samples failed to reach this level, including one each of the Arctic (17-0103-116R-FFPE), Africa-3 (V039) And Indian Subcontinent (V114) lineages with coverages of 58%, 59%, and 41%, respectively. A group of four FFPE samples of the *Lasiurus* bat type (98H-0337-84-FFPE, 98H-0338-5-FFPE, 98H-0341-128R-FFPE, and 98H-0342-90-FFPE) ranged from 34 to 58% genome coverage. Despite these lower values, these samples grouped within their respective clades with good bootstrap support. Moreover, accurate variant assignment of many samples was noted in several American Indigenous clades including those of the big brown bat (*Eptesicus fuscus*) host. Typing of the viruses associated with this host in the USA have identified four variants: EF-E1, EF-E2, EF-W1, and EF-W2 according to their range in the eastern and western parts of the country [34]. A Canadian study identified five distinct clades BB1 to BB5 [24] which correspond to the US variants thus: BB1 = EF-W2, BB2 = EF-E1, BB3, BB4, and BB5 = EF-E2. All RABV test samples from big brown bat hosts clustered according to their expected classification regardless of the sample type. In addition, test samples derived from *Myotis* bat hosts all clustered within the North American *Myotis* bat clade and were clearly delineated as expected into two subclades MYO I and MYO II [23].

### 3.5. Optimization of Viral Variant Analysis

Given that genome coverage and phylogenetic placement was clearly significantly impacted by the nature of the sequence used for reference-guided assembly, we explored the potential to maximize genome coverage and thereby reveal finely resolved epidemiological relationships of samples processed by the Ampliseq for Illumina approach. This was achieved for the Arctic test samples which were compared to a reference collection of previously described sequences representative of this lineage which segregates into four sub-lineages Arctic1 to Arctic4 [35,36]. The Arctic1 sub-lineage circulates only in the Canadian province of Ontario where it has evolved into four geographically restricted variants, ON1 to ON4 [37]. The Arctic2 sub-lineage has historically circulated in several northern countries, but few samples have been extensively investigated while Arctic4 is limited in range to Alaska. The Arctic3 sub-lineage is widely dispersed across the northern hemisphere and is currently the dominant type across Canada and Greenland where 18 distinct variants were previously recorded [35]. The Ampliseq data for the five unfixed and the eight FFPE samples of the Arctic lineage included in this study were reassembled using an Arctic lineage sequence as a reference (NT.1993.0669AFX). This same sample was indeed included as an unfixed sample in the study (93RABL0669). For unfixed samples, this approach improved sequence fidelity but had minimal impact on genome coverage while for the FFPE samples a significant improvement in fidelity and genome coverage was observed. The resulting assemblies were aligned with a set of sequences of the Arctic lineage representative of all the major genetic variants reported in Canada and neighboring northern countries (Appendix A) and this sequence set was used to generate an NJ tree (Figure 9).

This phylogeny clearly shows that selecting a reference sequence belonging to the Arctic lineage to assemble the Arctic lineage test sample genomes enables their association not only with the correct RABV lineage (Arctic) and sub-lineage (Arctic1 to Arctic4) but with the correct variant of that lineage with strong support. Thus, three test samples cluster within the most common ON2 variant of the distinctive Arctic1 sub-lineage, one test sample clusters within each of the sub-lineages Arctic2 and Arctic4, while the remaining eight test samples cluster with distinct variants of the Arctic3 sub-lineage, including Arctic3-2, Arctic3-4, and two closely related variants, Arctic3-17 and Arctic3-18, which do not resolve well in this tree. Seven of these 13 test samples had been sequenced previously, and in all cases the Ampliseq for Illumina-derived sequence clustered closely with the complete genome sequence and supported a similar variant designation. The biggest discrepancy was represented by sample 17-0663-169FFPE which was somewhat distant from its corresponding sample 2017NU0663AFX but still within the same variant clade.

## 4. Discussion

This study has demonstrated the value of an Ampliseq for Illumina panel to generate amplicons covering much of the viral genome for a wide variety of RABV variants. Parallel sequencing of these amplicons provided extensive viral genome coverage in many cases, thereby enabling detailed molecular epidemiological analysis. However, it became evident during the analysis of these reads that the selection of the reference sequence used in the reference-guided assembly was critical for optimizing the subsequent sequence analysis. While the source of the sample itself can often guide selection of an appropriate reference sequence, this may not always be the case. As in this study, widely divergent RABV lineages cocirculate in the Americas, and throughout the world there is the possibility of an introduction of non-native variants. It was therefore necessary to devise a means of using preliminary sequence data to predict a likely viral type that would inform the appropriate reference sequence to be used for subsequent assembly. Accordingly, a region of the L gene corresponding to bases 7396–8603 of the consensus RABV genome was employed. This region, which corresponds to bases 7362–8570 of the PV strain, encodes amino acids 649–1051 of the polymerase corresponding to the relatively conserved domains III and IV which contain critical functional motifs [38]. The high conservation of the genome flanking these domains resulted in relatively high coverage of this region by the Ampliseq for Illumina panel for virtually all samples except for some samples of the *Lasiurus* group. This genomic region, or portions thereof, was employed for BLAST analysis against the *Lyssavirus* rabies sequence collection in the NCBI GenBank database to identify the best match for each test sample and thereby predict its viral type. Despite the conserved nature of this region, there was adequate base variation to permit differentiation of most viral types thus facilitating further read assembly using an appropriate reference sequence. Using this optimization procedure, with few exceptions, most samples in this study generated a RABV genome coverage in the range of 70–95%. However, the BLAST analysis cannot be considered completely accurate; use of shorter sequences (<500 bases) or identity matches of <90% should be interpreted with caution as many RABV variants will yield a good match with these criteria. In addition, rare variants that are poorly represented in the NCBI database will likely be misidentified in terms of their viral type. Such a situation may be responsible for the erroneous typing of the Mexican sample (V854) by the BLAST analysis. Additionally, it complicated the precise typing of the two samples from Puerto Rico for which good references were not available. Indeed, it was noted that a few samples of the Cosmopolitan lineage were not precisely typed by BLAST analysis presumably due to the limited genetic divergence within this lineage, but all samples of this lineage yielded good coverage in assembly A1 using the consensus RABV sequence as reference.

While the current panel, which was especially focused on the main variants circulating in the Americas, performed well for many variants it was less successful with some samples of the *Lasiurus* group for which four FFPE samples failed to meet a threshold genome coverage of 60% and another two FFPE samples yielded coverage just above 60%. Reasons for this reduced coverage could include either low viral titer in these samples, poor match of some of the panel primers to their sequence targets, or the RNA from these samples may be of particularly low quality. It should be mentioned that these six FFPE samples originated from two submissions that were processed in triplicate during the fixation. The RT-qPCR Ct values for all six FFPE samples were in line with those of other FFPE samples (ranging from 14.39 to 17.47) arguing against a particularly low viral titer. Interestingly, two unfixed *Lasiurus* samples from the USA (V026 and V231 having Ct values of 7.26 and 7.73, respectively) yielded genome coverage values of 63% and 83% suggesting a rather poor correlation between viral load and genome coverage. Overall, the replicate samples included in the FFPE group generally yielded similar results. All real-time RT-qPCR Ct values were within 1.5 Ct except for one set for which the range was 2.5 Ct. Furthermore, replicate samples yielded consistent trends with respect to genome coverage using different reference sequences and they performed similarly in the phylogenetic analysis. This demonstrates the consistency of the method as currently performed.

Within the unfixed sample group, evidence that higher viral loads resulted in higher genome coverage was mixed. This appeared to be the case for the two unfixed Africa-3 samples, V039 and V050, for which Ct values were 15.81 and 8.59 and genome coverage was 59% and 90%, respectively. However, of the three Indian Subcontinent samples V114 had the highest viral load (Ct of 9.89 compared to 14.53 and 14.46 for V113 and V115, respectively) but the lowest genome coverage (41% compared to 89% and 95%). Clearly factors other than viral load impacted the final outcome of these studies. Future work to further optimize this methodology would benefit from studies to explore the effect of tweaking the primer panel to best suit the viral variants of specific regions and to determine the optimal starting amount of RNA template based on the viral load (for example using RT-qPCR analysis) given the variability of the viral RNA template in a sample. Furthermore, it should be acknowledged that the present study was performed with the prior knowledge of the viral type of the test samples thus potentially introducing bias in the analytical approach. Verification of this methodology would benefit from studies using double blinded panels to check for precision and accuracy of inferred viral types.

While the Ampliseq for Illumina method described in this report can be applied to RNA recovered from either unfixed or FFPE tissue samples, its greatest value is in its application to the latter tissue type. To date, FFPE material has remained refractory to detailed molecular epidemiological analyses of RABVs due to challenges in generating extensive sequence information from this source. Preliminary analyses also suggest that this method may be a useful tool for the detailed analysis of material stored on solid surfaces such as Whatman FTA cards. Future iterations of this method will facilitate the analysis of archived tissue samples, including FFPE material, thereby furthering our knowledge of the historical spread of rabies.

## Figures and Tables

**Figure 1 viruses-14-02241-f001:**
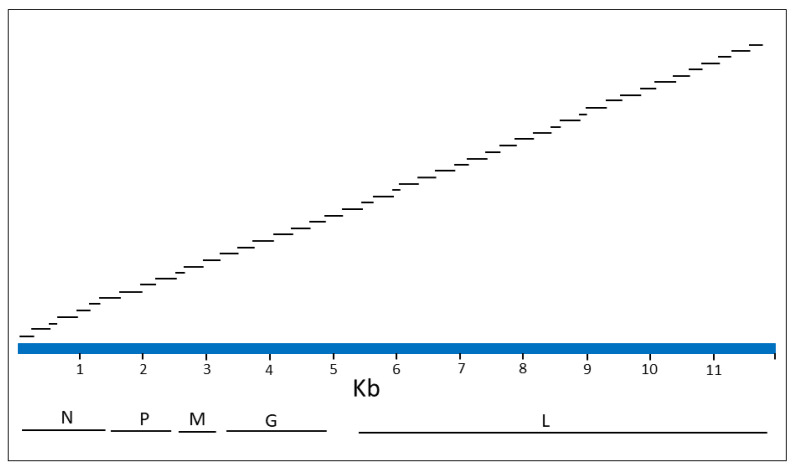
A schematic of the Ampliseq for Illumina primer panel. The 11.967 Kb RABV consensus sequence genome (positive sense) is represented by a blue bar above which the locations targeted by the 47 panel primer pairs are shown. For each of the 47 amplicons only the internal sequence is illustrated; amplicon size is 50 bp longer with inclusion of the primers. Illustrated below the genome are the locations of the five viral genes that encode products as follows: N, nucleoprotein; P, phosphoprotein; M, matrix protein; G, glycoprotein; L, polymerase.

**Figure 2 viruses-14-02241-f002:**
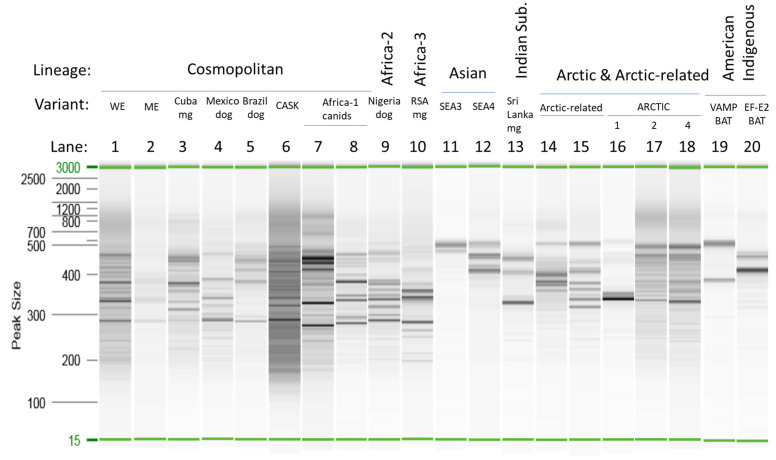
Amplicon profiles of sample aliquots from the cleaned Ampliseq for Illumina library as analyzed by a QIAxcel system. A 15–3000 bp reference marker was included in each run. Lanes 1–20 represent the unfixed samples as follows: 1, V285; 2, V661; 3, V1061; 4. V682; 5, V904; 6, V648; 7, V671; 8, V1453; 9, V463; 10, V050; 11, V1145; 12, V1375; 13, V114; 14, V704; 15, V737; 16, 01RABN00053; 17, V809; 18, V804; 19, V982; and 20, 72RABL03675. Variants were identified as indicated in Appendix A. Abbreviations: mg, mongoose; Indian Sub., Indian Subcontinent; RSA, Republic of South Africa.

**Figure 3 viruses-14-02241-f003:**
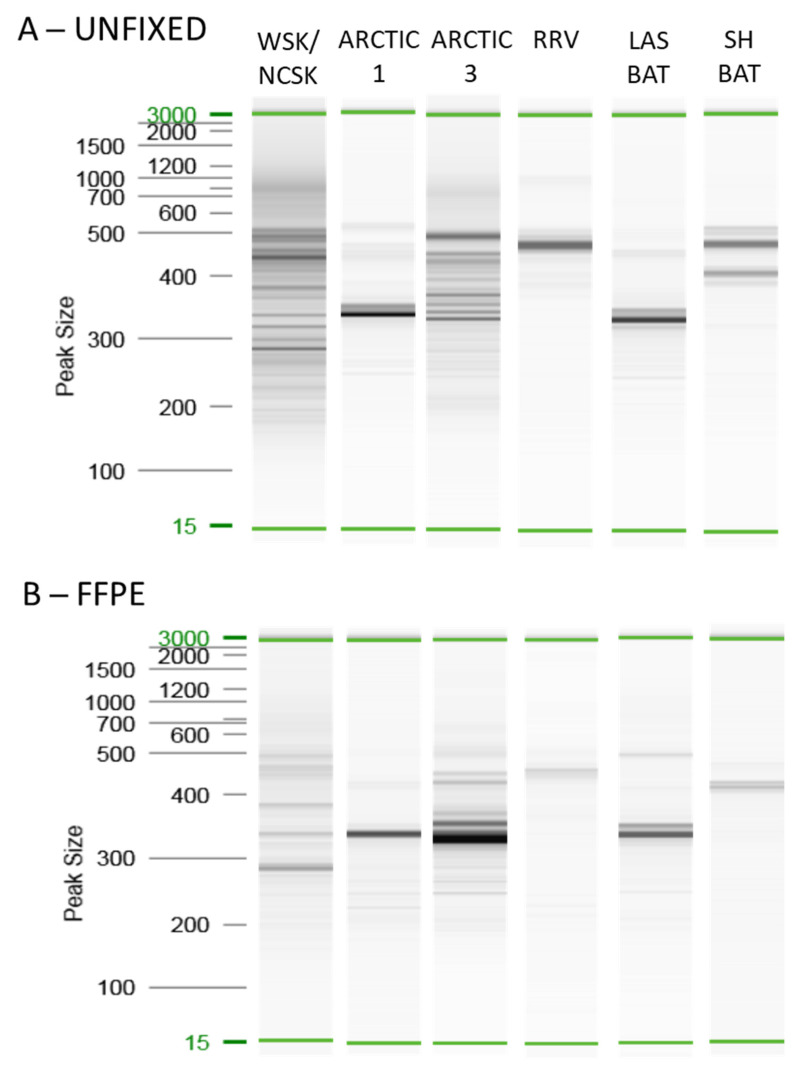
A comparison of amplicon profiles generated by the Ampliseq for Illumina protocol for fixed and FFPE samples of the same RABV variant. The cleaned sequencing library was analyzed by a QIAxcel system with the inclusion of a 15–3000 bp reference marker in each run. Results for unfixed samples (**A**) and FFPE samples (**B**) are shown for the following RABV variants: Western skunk/North Central skunk (WSK/NCSK) 04RABL00965, 92RABL01670; Arctic1 01RABN00053, 07RABN06558; Arctic3 91RABN05406, 17RABN00630; Mid-Atlantic raccoon (RRV) ME.2014.0197, 18RABN01852; *Lasiurus* bat (LAS BAT) V231, 94RABN04952; and Silver-haired bat (SH BAT) V077, 93RABL01950.

**Figure 4 viruses-14-02241-f004:**
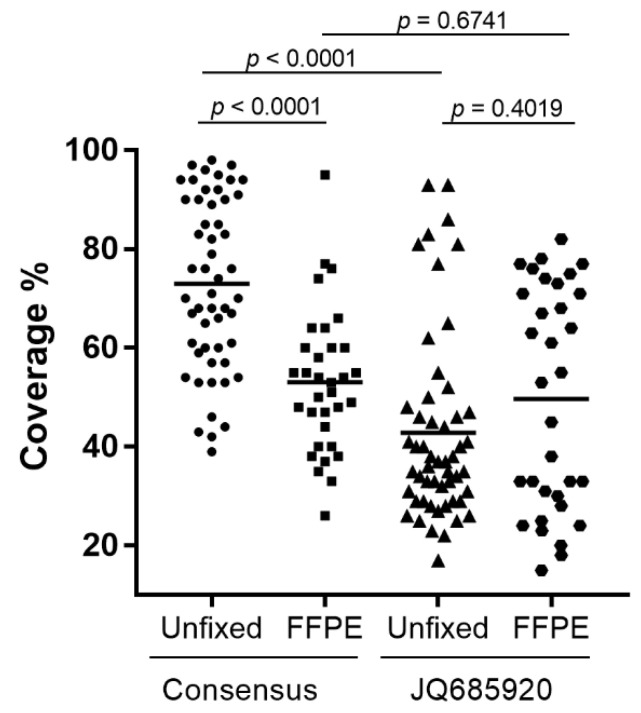
Scatter plots of the percentage coverage of the rabies virus genome of Ampliseq reads for all samples using either the consensus (Appendix A) or JQ685920 sequences for reference-guided assembly. Unfixed and FFPE samples are illustrated separately. Horizontal bars indicate the average value in each group and the calculated *p*-values between each group are shown above the diagram.

**Figure 5 viruses-14-02241-f005:**
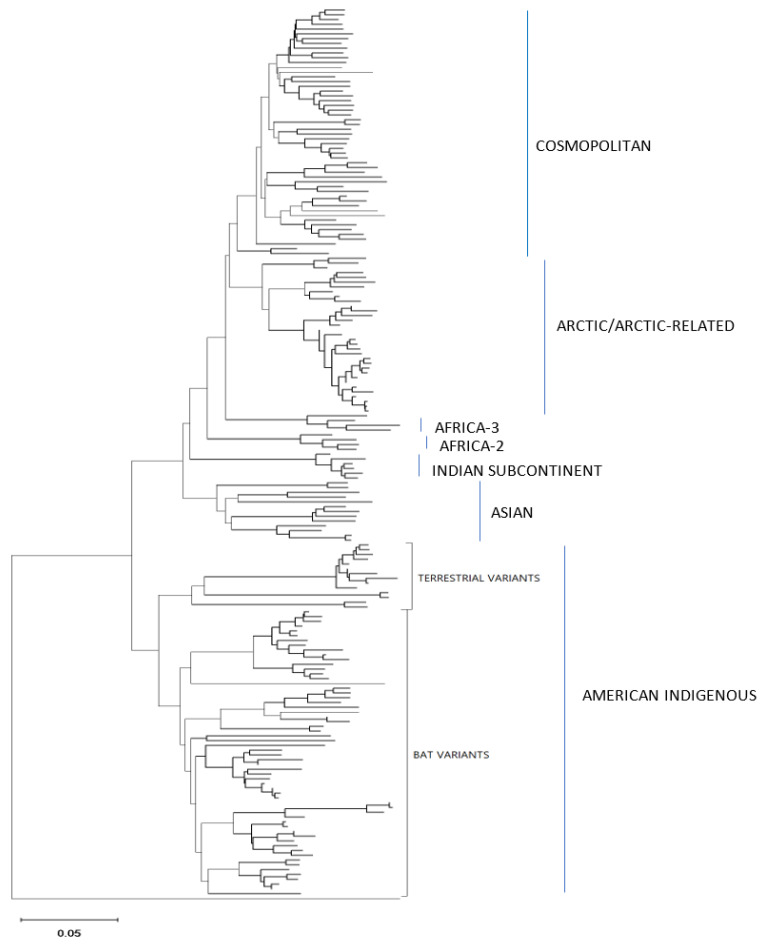
Neighbor joining tree of 86 RABV samples analyzed by the Ampliseq for Illumina protocol and 100 representative RABV genomes. Each sample genome was reconstructed through reference-guided assembly of the Ampliseq reads using one of eight reference sequences and a final alignment of 11,777 positions was generated. During the tree construction ambiguous positions and deletions were removed using a pairwise deletion option. An Australian bat lyssavirus (ABLV) sample was used as an outgroup. Lineages and variants are identified to the right of the tree.

**Figure 6 viruses-14-02241-f006:**
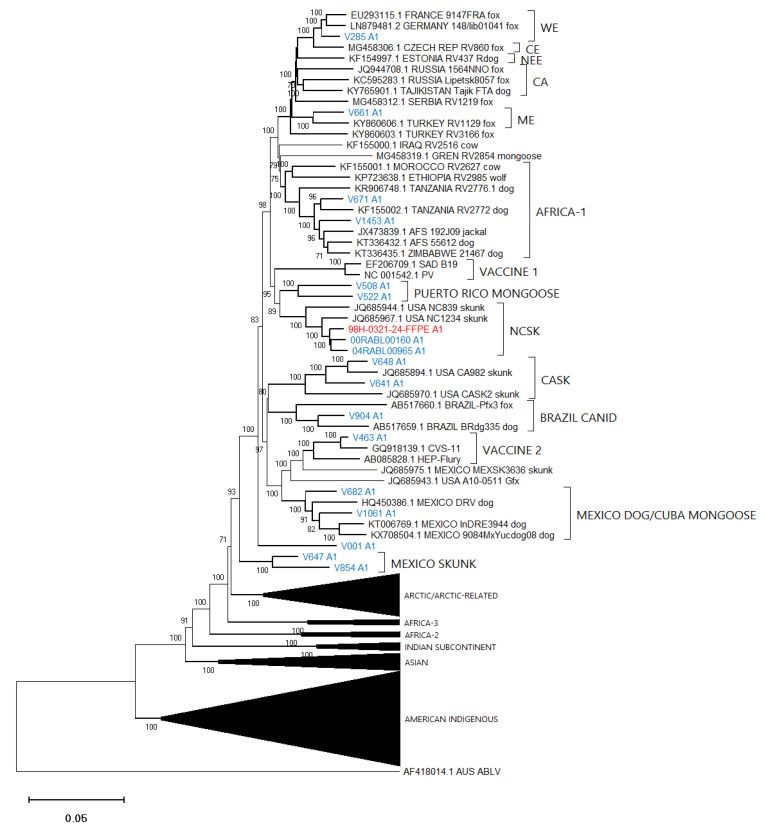
Neighbor joining tree of 86 RABV test samples and 100 representative RABV genomes showing details of the Cosmopolitan lineage. Sample genomes were reconstructed through reference-guided assembly of the Ampliseq reads using eight different references as indicated by the A1 to A8 suffix following the taxon name. Sequences generated from unfixed samples are shown in blue while those from FFPE samples are in red. Numbers at nodes indicate bootstrap values ≥70%. Lineages and variants are identified to the right of the tree.

**Figure 7 viruses-14-02241-f007:**
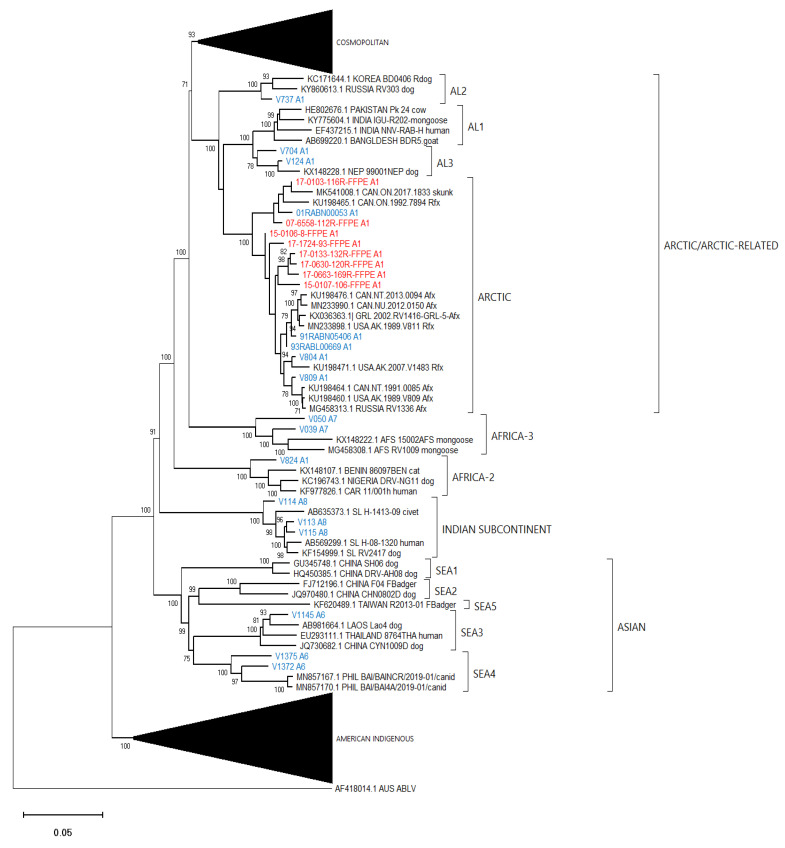
Neighbor joining tree of 86 RABV test samples and 100 representative RABV genomes showing details of the Arctic and Arctic-related, Africa-2, Africa-3, Asian and Indian Subcontinent lineages. Sample genomes were reconstructed through reference-guided assembly of the Ampliseq reads using eight different references as indicated by the A1 to A8 suffix following the taxon name. Sequences generated from unfixed samples are shown in blue while those from FFPE samples are in red. Numbers at nodes indicate bootstrap values ≥70%. Lineages and variants are identified to the right of the tree.

**Figure 8 viruses-14-02241-f008:**
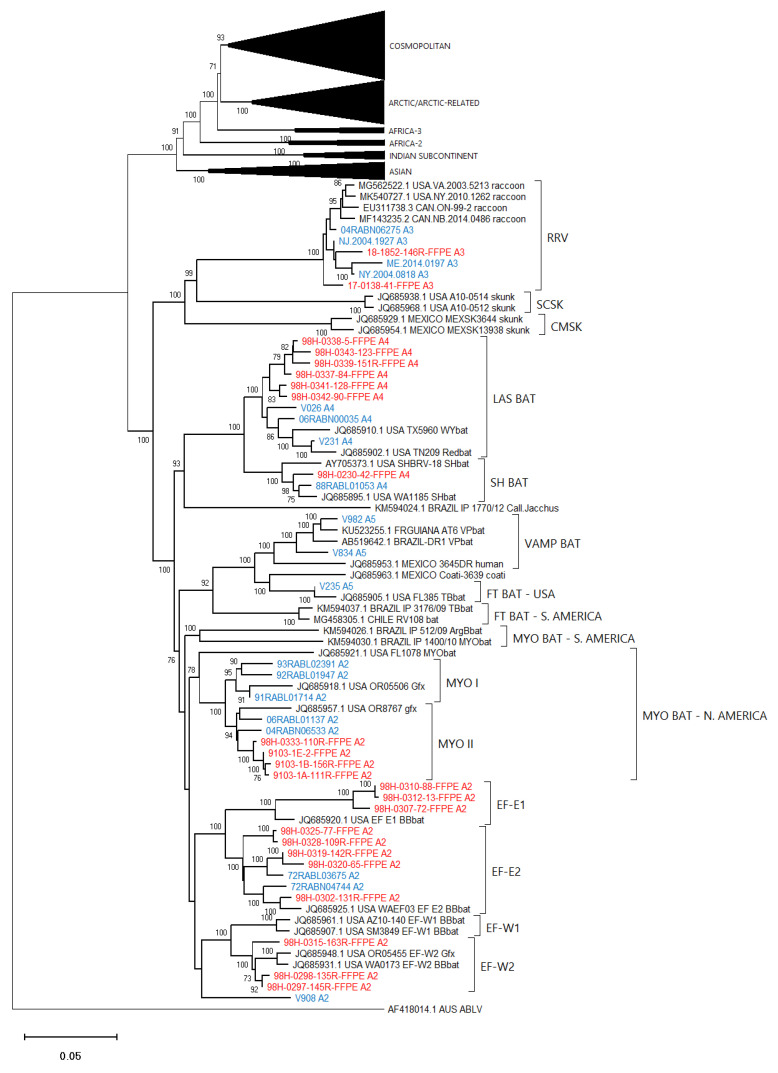
Neighbor joining tree of 86 RABV test samples and 100 representative RABV genomes showing details of the American Indigenous lineage. Sample genomes were reconstructed through reference-guided assembly of the Ampliseq reads using eight different references as indicated by the A1 to A8 suffix following the taxon name. Sequences generated from unfixed samples are shown in blue while those from FFPE samples are in red. Numbers at nodes indicate bootstrap values ≥70%. Lineages and variants are identified to the right of the tree.

**Figure 9 viruses-14-02241-f009:**
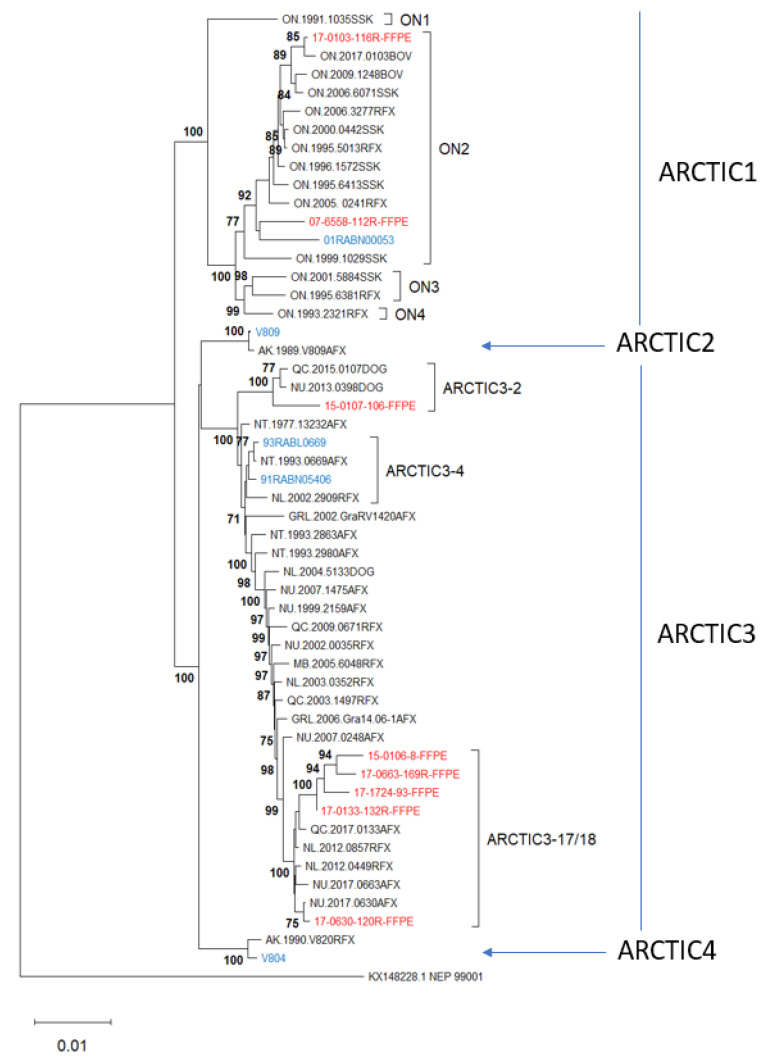
A Neighbor joining tree of 13 Ampliseq for Illumina samples and 39 RABVs representative of the Arctic lineage, as well as the Arctic-related outlier sample NEP99001 (AL3). The Ampliseq for Illumina samples were assembled using sample NT.1993.0669AFX (accession #MN233954) as a reference and then incorporated into the RABV whole genome alignment of the representative samples, details of which have been published [35,36]. The final dataset comprised 11,800 positions and was subject to phylogenetic analysis as described. Unfixed test samples are shown in blue and FFPE samples in red. Numbers at nodes indicate bootstrap values ≥70%. Clades are identified to the right of the tree.

**Table 1 viruses-14-02241-t001:** Species assignment of raw Illumina reads by Kraken2.

Sample Name	01RABN00053	72RABL03675	V1145	V1375	V804	V809	V982
Host species	*Mephitis mephitis*	*Eptesicus fuscus*	*Canis familiaris*	*Canis familiaris*	*Vulpes vulpes*	*Vulpes lagopus*	*Bos taurus*
Number of raw reads	225,275	433,412	201,030	224,329	254,338	312,591	416,666
Classified reads (%)	99.55	99.40	98.98	98.20	99.26	99.49	98.46
Unclassified reads (%)	0.45	0.60	1.02	1.80	0.74	0.51	1.54
Viral reads (%)	99.49	99.38	98.08	97.85	99.24	99.49	98.42
Chordate reads (%)	0.06	0.02	0.91	0.35	0.01	0.00	0.04
Canidae reads (%)	0.00	0.00	0.87	0.19	0.00	0.00	0.01
Mephitidae reads (%)	0.04	0.00	0.02	0.01	0.00	0.00	0.00
Bovidae reads (%)	0.00	0.00	0.00	0.00	0.00	0.00	0.01
Chiroptera reads (%)	0.00	0.00	0.00	0.00	0.00	0.00	0.00
Comments	99 reads assigned to *Mephitidae*	4 reads assigned to *Eptesicus fuscus*	1746 reads assigned to *Canidae*	424 reads assigned to *Canidae*	8 reads assigned to *Vulpes* sp.	3 reads assigned to *Vulpes* sp.	21 reads assigned to *Bos taurus*

## Data Availability

Aligned sequence data are presented in the Appendix A.

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
