# Peer review of "Ampliseq for Illumina Technology Enables Detailed Molecular Epidemiology of Rabies Lyssaviruses from Infected Formalin-Fixed Paraffin-Embedded Tissues"

_viruses, 2022, doi:10.3390/v14102241_

Round 1

Reviewer 1 Report

The research manuscript entitled “Ampliseq for Illumina technology enables detailed molecular epidemiology or rabies lyssaviruses from infected formalin fixed paraffin-embedded tissues” by Nadin-Davis et. al., entails the implementation of a high throughput method for the robust inference of the RABV variant and/or linage involved in rabies cases where tissues of the potentially rabid human or animal were formalin fixed and paraffin embedded for subsequent diagnosis by immune-histochemical or pathology studies. To this end, authors implemented a multiplexed designed developed by Illumina so-called “Ampliseq” to generate at once 47 whole genome overlapping amplicons (average sizes between 152-377 bp expected or 200-400 bp obtained), which in theory, would cover for up to 98.16% of the entire RABV genome. Authors designed a robust validation strategy for proof of concept, as well as implemented a standard analytical pipeline encompassing verification of sequence coverage, assessment, sorting (proportion of viral vs host sequences from total raw reads) of quality reads, and guiding assembling using different reference sequences. The Ampliseq primer sets implemented by authors, allowed them to selectively enriched RABV amplicons as showed by Kraken2 analysis. Surprisingly, the overall genomic coverage was limited in several samples. Unfortunately, authors report highly discrepant genome coverages across assembled genomes using different reference sequences, which in turn, yielded inconsistent phylogenetic reconstructions for the same sequence sets assembled. Nonetheless, knowing a priori (blasting or by preliminary analyses) what the closest relative of the tested sequences could be, authors showed whole genome sequence coverage could somewhat be improved. In the long run, authors were able to identify the lineage involved in every tested sequence, however with limited precision and accuracy.

Although, the methodological approach may be promising to address current limitations to provide RABV lineage/variant identification in formalin fixed-paraffin embedded tissues, the need for using multiple reference sequences to optimize sequencing assembling brings outstanding challenges for inexperienced technicians. At the end of the day such challenges may severely jeopardize reproductivity (precision) and accuracy for lineage/variant inference, which may negatively impact control and prevention strategies downstream.

I am confident that the analytical pipeline can be significantly improved, which requires tailored assembling approaches, as well as customization of the multiple sequence alignment for tree reconstruction. Once the precision and accuracy hurdles inherent to this methodology are properly handle, the implemented methodology would be great inform relevant epidemiological details on the potential sources for rabies exposure, such as, inferring the reservoir host involved, geographic range of the involved RABV lineage, sublineage or variant. Lineage inference can ultimately guide epidemiologist to search for details on the history of exposure in any given case, inform and tailor field rabies control and containment measures based on the reservoir inferred (e.g., massive vaccination to rabies reservoir host involved), provide timely risk assessments and post exposure prophylaxis to first responders and care givers associated with the case, as well as launching informational campaigns to alert people on the potential rabies exposure sources in the region.

Overall, authors conducted an extraordinary research effort to cover an extremely relevant technical gap for the typing of RABV in formalin fixed paraffin embedded tissues by implementing a cutting age technology with a robust experimental design and sounded analytical approaches. For those reasons I strongly recommend publication of this article. However, is recommend that authors address some analytical weaknesses to improve accuracy and precision in their approach by either suggesting or executing alternative analyses before releasing a final publication.

1)      Please, provide better resolution on Figures 5 and 6. Taxon names and bootstrap values are illegible, even when zooming in. In fact, zooming in, makes the figure’s taxon names less legible. Thus, it is not possible to follow individual taxon across discrepant tree topologies generated. I wonder if figures can be sent as separate files. Alternatively, I would try to depict trees on single pages. Some journals provide links to gain access to higher resolution images.

I would also suggest depicting taxa names on trees, for fixed and unfixed samples, with different colors to facilitate follow up over different tree topologies.

2)      I strongly recommend, seeking alternatives to improve sequence assembling and edition, so sequence multiple alignments get optimal for tree reconstruction. I think that is the main cause for the lack for consistency in tree topologies for the same date sets (obtained with different ref seqs) and weak clustering patterns (low bootstrap values). Sequence sets presenting a highly variable range of complete genome coverage (as shown in sequences assemblies presented in the supplementary materials) generate less than ideal multiple sequence alignments that result in highly discrepant tree topologies, which complicates a robust consistent inference of RABV lineages, sub lineages and variants. Although, gaps can be removed during or before the execution of a neighbor-joining tree reconstruction, the algorithm has problems to reconstruct phylogenies with sequences of different lengths producing trees with very poor cluster support values. If authors would like to proceed with tree reconstructions with no further edition of the alignments. I would then recommend using Bayesian inference rather than neighbor-joining, given that Bayes implemented in Beast offers an efficient a highly robust way to account for sequences of different lengths.

Conversely, some authors have resolved this problem by concatenating amino acid coding regions only (Troupin C, Dacheux L, Tanguy M, Sabeta C, Blanc H, Bouchier C, Vignuzzi M, Duchene S, Holmes EC, Bourhy H. Large-Scale Phylogenomic Analysis Reveals the Complex Evolutionary History of Rabies Virus in Multiple Carnivore Hosts. PLoS Pathog. 2016 Dec 15;12(12):e1006041. doi: 10.1371/journal.ppat.1006041. PMID: 27977811; PMCID: PMC5158080.), by removing leader, non-coding (intergenic regions), and subsequently identify coding regions (post assembling) before embarking in multiple sequence alignments and phylogenetic reconstructions. Sequences with less than 90% coverage may even lack several complete structural genes (as the assembling alignments shown in supplementary figures for this research), resulting in multiple sequence alignments with several sequence gaps all over the multiple “complete genome”) sequence alignment (most tested sequence were highly incomplete genome). Alternatively, I would try to first identify what genes or coding regions are 100% covered in most sequences and then reconstruct trees with the concatenated genes or coding regions present in all or most tested sequences. In the high variability in coverage still does not allow for this. Sequences with highly discrepant coverages should be handled individually or by groups in sake to improve precision, as well as accuracy in the phylogenetic inferences of lineages and variants, as showed for the artic analysis.

3)      Another option, proposed by authors would be to try a customize guided assembling, as further phylogenetic analysis as shown for the artic group. However, this entails to know a priori what lineages are likely occurring in the samples tested.  To this end using, author can suggest the search for the closest relative or seeking the best BLAST match (>90% identity), for each tested sequence using identified robust genes for phylogenetic inference of a RABV lineage or variant, such as partial or complete N gene sequences. This best match can, then be used to tailor the guided sequence assembling. If sequence coverage in not improved significantly then we can be dealing with problems related to the yield of amplicons. Likely because concentration of template has to be optimized as authors discussed; or generation of shorter amplicons should be pursued; or the affinity of the primer design is very poor that I believe is the less likely of the possibilities given the evidence presented in the amplicon profiles.

4)      I perceive an over optimism in authors when discussing about the accuracy and precision of their results to inferred lineage/variants in the samples tested, particularly when their results highlight otherwise.  One of their main conclusions claims they could infer accurately most of the lineages and variants involved in their research, when the data shows evident complications (just by the topology discrepancies shown in Figures 5 and 6) that may jeopardize a right judgment for the naïve eye. Authors should acknowledge that the lineage of all samples studied was known a priori, which inflicted certain bias at the moment of interpreting the phylogenetic inferences. If this information were not available, tree topology interpretations and therefore lineage/sub-lineage/variant inferences would have been extremely challenging, most likely incorrect. I think to move validation forward the approach suggested here should be further tested with double blinded panels to check for precision and accuracy on the inferences for lineages/sub-lineages/variants.

Although, implementation of an Ampliseq seems promising, necessary, and timely. The implementation of such approach still requires significant further improvement efforts along the entire analytical pipeline starting from sequence assembling, sequence post assembly edition to improve precision and accuracy of linage/variant inference. Perhaps, also some primer alternatives should be explored to see if coverage can be improved. Particularly, trying shorter amplicons in the order of 150-200 bp and play more with template concentrations to see if amplicon yield can be significantly improved. I suggest authors should be more emphatic to highlight limitations of their approach to pave the way forward more securely towards the implementation of necessary improvements.

Minor observations

Line 234 I would suggest adding in parenthesis after “panel failed to support amplification” (due to differences in primer affinities across different variants and/or lineages).

After lines 236-237 I think there could be a third hypothesis that may be considered, which would account for the availability of longer and higher concentrations of viral -SSRNA templates in unfixed tissues that in turn increases the probability for the yield of longer amplicons. Meanwhile, formalin fixed tissues may only have available short length -SSRNA templates, due to nucleic acid fragmentation during the fixation procedure, thereby yielding higher concentrations of shorter amplicons as observed in amplicons patterns on Figures 2 and 3. The species assignment analysis better supports this hypothesis over the one suggesting amplification of longer length could predominantly come from host DNA. Along the same lines poor sequence coverage in several samples may indicate less than ideal concentration of -SSRNA templates during the multiplex amplification, which need to be highlight by authors with more emphasis as an impediment to generate homogenous genome coverages over 90% in all samples tested.

Lines 244-245 authors indicate that samples with Ct values under 20 yielded fainted amplicons profiles. This could be a clear indication that some samples may contain less than ideal quantities of -SSRNA compromising ideal whole genome coverage. Ideally, higher that 90%.

Line 311 I would suggest adding the word intergenic or non-coding region after G-L., “highly divergent G-L region” for “highly divergent G-L intergenic region” to make emphasis this is a no amino acid coding region.

Lines 364-373 present a remarkable finding. However, it is concerning that its placement in the overall tree topologies is not consistent throughout phylogenetic reconstructions and differs for that commonly accepted in the field. These indicates concerning analytical noises likely introduced for the use of less-than-ideal multiple sequence alignments, which could be due to the presence of radically different sequence gaps generated in the same sample tested during the guided assembling using different rabies reference sequences.

Lines 444-447 I think this sentence is somehow confusing since it is not quite consistent with the overall results. Although the implemented Ampliseq Illumina approach demonstrated an ideal viral amplicon enrichment (over 90%), that was not sufficient to attain ideal genomic coverages (an average of >50%), which compromise and confounded downstream lineage/variant inferences. I would current this statement accordingly.

Lines 454-456 I think this would have been more accurate is samples tested would have been analyzed separately rather than in blocks. Aligning samples with multiple gaps across their genomes imposes severe analytical challenges for phylogenetic reconstructions. Neighbor-joining reconstruction should only be used for gapless perfectly aligned multiple sequences alignments. Otherwise, more sophisticated reconstruction algorithms such as the Bayesian one implemented in the Beast package should be attempted.

Lines 467-473 I think this paragraph should be rephrased to improve accuracy. Results presented throughout this investigation demonstrate that the diversity of RABV present in the Western Hemisphere requires the use of tailored sequence assembling by using reference sequences pertaining to the same lineage/variant expected, which complicates and limits the optimization and utilization of this approach as a high throughput technology, unless the analytical pipeline gets meticulously improved.

Author Response

Our responses to the reviewer’s comments are provided below in bold. Line numbers refer to the revised version with tracked changes. We would like to thank Reviewer 1 for insightful comments which led to a significant modification to the analytical pipeline employed in this study thus substantially improving the accuracy of the analysis and greatly augmenting the quality of the manuscript.

Reviewer 1

The research manuscript entitled “Ampliseq for Illumina technology enables detailed molecular epidemiology or rabies lyssaviruses from infected formalin fixed paraffin-embedded tissues” by Nadin-Davis et. al., entails the implementation of a high throughput method for the robust inference of the RABV variant and/or linage involved in rabies cases where tissues of the potentially rabid human or animal were formalin fixed and paraffin embedded for subsequent diagnosis by immune-histochemical or pathology studies. To this end, authors implemented a multiplexed designed developed by Illumina so-called “Ampliseq” to generate at once 47whole genome overlapping amplicons (average sizes between152-377 bp expected or 200-400 bp obtained), which in theory, would cover for up to 98.16% of the entire RABV genome. Authors designed a robust validation strategy for proof of concept, as well as implemented a standard analytical pipeline encompassing verification of sequence coverage, assessment, sorting (proportion of viral vs host sequences from total raw reads) of quality reads, and guiding assembling using different reference sequences. The Ampliseq primer sets implemented by authors, allowed them to selectively enriched RABV amplicons as showed by Kraken2analysis. Surprisingly, the overall genomic coverage was limited in several samples. Unfortunately, authors report highly discrepant genome coverages across assembled genomes using different reference sequences, which in turn, yielded inconsistent phylogenetic reconstructions for the same sequence sets assembled. Nonetheless, knowing a priori (blasting or by preliminary analyses) what the closest relative of the tested sequences could be, authors showed whole genome sequence coverage could somewhat be improved. In the long run, authors were able to identify the lineage involved in every tested sequence, however with limited precision and accuracy.

Although, the methodological approach may be promising to address current limitations to provide RABV lineage/variant identification in formalin fixed-paraffin embedded tissues, the need for using multiple reference sequences to optimize sequencing assembling brings outstanding challenges for inexperienced technicians. At the end of the day such challenges may severely jeopardize reproductivity (precision) and accuracy for lineage/variant inference, which may negatively impact control and prevention strategies downstream.

I am confident that the analytical pipeline can be significantly improved, which requires tailored assembling approaches, as well as customization of the multiple sequence alignment for tree reconstruction. Once the precision and accuracy hurdles inherent to this methodology are properly handle, the implemented methodology would be great inform relevant epidemiological details on the potential sources for rabies exposure, such as, inferring the reservoir host involved, geographic range of the involved RABV lineage, sublineage or variant. Lineage inference can ultimately guide epidemiologist to search for details on the history of exposure in any given case, inform and tailor field rabies control and containment measures based on the reservoir inferred(e.g., massive vaccination to rabies reservoir host involved),provide timely risk assessments and post exposure prophylaxis to first responders and care givers associated with the case, as well as launching informational campaigns to alert people on the potential rabies exposure sources in the region.

Overall, authors conducted an extraordinary research effort to cover an extremely relevant technical gap for the typing of RABV in formalin fixed paraffin embedded tissues by implementing a cutting edge technology with a robust experimental design and sounded analytical approaches. For those reasons I strongly recommend publication of this article. However, is recommend that authors address some analytical weaknesses to improve accuracy and precision in their approach by either suggesting or executing alternative analyses before releasing a final publication.

Response

We thank the reviewer for the informed support given to this methodology and the acknowledgement of its potential benefit to studies that seek to further our knowledge of rabies spread and elicit its control. We acknowledge the limitations of the Ampliseq sequence analysis that was presented in the initial report, and, in light of the reviewer’s excellent suggestions, we have undertaken extensive steps to correct these deficiencies as reported below. As a result, the manuscript has been extensively revised. 

1)

Please, provide better resolution on Figures 5 and 6. Taxon names and bootstrap values are illegible, even when zooming in. In fact, zooming in, makes the figure’s taxon names less legible. Thus, it is not possible to follow individual taxon across discrepant tree topologies generated. I wonder if figures can be sent as separate files. Alternatively, I would try to depict trees on single pages. Some journals provide links to gain access to higher resolution images.

I would also suggest depicting taxa names on trees, for fixed and unfixed samples, with different colors to facilitate follow up over different tree topologies.

Response

Figures 5 and 6 have been replaced with a single new phylogeny generated using customized Ampliseq assemblies optimized for each RABV lineage (see details in response to item 3). The entire tree is shown (topology only) as Figure 5 and portions of this tree are detailed in Figures 6-8. These subsequent trees permit easy visualization of individual taxon names and nodal bootstrap values. In addition, test samples are colored for easy identification – unfixed samples in blue and FFPE samples in red.

2)

I strongly recommend, seeking alternatives to improve sequence assembling and edition, so sequence multiple alignments get optimal for tree reconstruction. I think that is the main cause for the lack for consistency in tree topologies for the same date sets (obtained with different ref seqs) and weak clustering patterns (low bootstrap values). Sequence sets presenting a highly variable range of complete genome coverage (as shown in sequences assemblies presented in the supplementary materials) generate less than ideal multiple sequence alignments that result in highly discrepant tree topologies, which complicates a robust consistent inference of RABV lineages, sub lineages and variants. Although, gaps can be removed during or before the execution of a neighbor-joining tree reconstruction, the algorithm has problems to reconstruct phylogenies with sequences of different lengths producing trees with very poor cluster support values. If authors would like to proceed with tree reconstructions with no further edition of the alignments. I would then recommend using Bayesian inference rather than neighbor-joining, given that Bayes implemented in Beast offers an efficient a highly robust way to account for sequences of different lengths.

Conversely, some authors have resolved this problem by concatenating amino acid coding regions only (

Troupin C, DacheuxL, Tanguy M, Sabeta C, Blanc H, Bouchier C, Vignuzzi M, DucheneS, Holmes EC, Bourhy H. Large-Scale Phylogenomic AnalysisReveals the Complex Evolutionary History of Rabies Virus inMultiple Carnivore Hosts. PLoS Pathog. 2016 Dec15;12(12):e1006041. doi: 10.1371/journal.ppat.1006041. PMID:27977811; PMCID: PMC5158080.),

by removing leader, non-coding (intergenic regions), and subsequently identify coding regions (post assembling) before embarking in multiple sequence alignments and phylogenetic reconstructions. Sequences with less than 90% coverage may even lack several complete structural genes (as the assembling alignments shown in supplementary figures for this research), resulting in multiple sequence alignments with several sequence gaps all over the multiple “complete genome”) sequence alignment (most tested sequence were highly incomplete genome). Alternatively, I would try to first identify what genes or coding regions are 100% covered in most sequences and then reconstruct trees with the concatenated genes or coding regions present in all or most tested sequences. In the high variability in coverage still does not allow for this. Sequences with highly discrepant coverages should be handled individually or by groups in sake to improve precision, as well as accuracy in the phylogenetic inferences of lineages and variants, as showed for the artic analysis.

Response

A detailed account of the revised analytical pipeline which takes into account all reviewer’s suggestions is given below item 3.

3)

Another option, proposed by authors would be to try a customize guided assembling, as further phylogenetic analysis as shown for the artic group. However, this entails to know a priori what lineages are likely occurring in the samples tested.

To this end using, author can suggest the search for the closest relative or seeking the best BLAST match (>90% identity), for each tested sequence using identified robust genes for phylogenetic inference of a RABV lineage or variant, such as partial or complete N gene sequences. This best match can, then be used to tailor the guided sequence assembling. If sequence coverage in not improved significantly then we can be dealing with problems related to the yield of amplicons. Likely because concentration of template has to be optimized as authors discussed; or generation of shorter amplicons should be pursued; or the affinity of the primer design is very poor that I believe is the less likely of the possibilities given the evidence presented in the amplicon profiles.

Response

While exploring alternative pipeline options for Ampliseq data assembly, it was noticed that most samples yielded good coverage across a conserved region of the L gene after reference-guided assembly using the original reference sequence presented in Figure S1. Accordingly, we employed sequence obtained from this region for each test sample in a BLAST analysis against the complete rabies lyssavirus dataset available in GenBank to identify the closest sequence match. This process identified a putative viral type for each sample thus informing a more customized assembly process in which samples were reassembled using reference sequences appropriate for the lineage to which they were provisionally assigned. This procedure enabled higher levels of genome coverage to be attained for all samples; with a few exceptions many reached a value > 80%. Using these assemblies, performed with different reference sequences, a single gapped alignment was generated and used for tree generation. As illustrated in Figures 6-8, a neighbor joining analysis of this alignment yielded a highly robust tree with bootstrap values of 100% at many nodes. Clearly the issue with the previous trees was a problem in gaining sufficient genome coverage when samples were assembled using inappropriate references. This assembly pipeline provides a standard method for using Ampliseq sequence data to accurately identify the lineage of a viral sample with further customized assembly and phylogenetic analysis enabling detailed viral typing of the sample as shown with the Arctic data set (Figure 9).

4)

I perceive an over optimism in authors when discussing about the accuracy and precision of their results to inferred lineage/variants in the samples tested, particularly when their results highlight otherwise.

One of their main conclusions claims they could infer accurately most of the lineages and variants involved in their research, when the data shows evident complications (just by the topology discrepancies shown in Figures 5 and 6) that may jeopardize a right judgment for the naïve eye. Authors should acknowledge that the lineage of all samples studied was known a priori, which inflicted certain bias at the moment of interpreting the phylogenetic inferences. If this information were not available, tree topology interpretations and therefore lineage/sub-lineage/variant inferences would have been extremely challenging, most likely incorrect. I think to move validation forward the approach suggested here should be further tested with double blinded panels to check for precision and accuracy on the inferences for lineages/sub-lineages/variants.

Response

The revised data analysis pipeline presented in this revision enables an accurate identification of the viral lineage and variant of virtually all samples examined. In the discussion we acknowledge that the studies performed in this report were undertaken using previously typed samples and thus some bias in the interpretation was unavoidable and the reviewer’s suggestion that double blinded panels be used to verify this methodology has been added (lines 651 to 655).

Although, implementation of an Ampliseq seems promising, necessary, and timely. The implementation of such approach still requires significant further improvement efforts along the entire analytical pipeline starting from sequence assembling, sequence post assembly edition to improve precision and accuracy of linage/variant inference. Perhaps, also some primer alternatives should be explored to see if coverage can be improved. Particularly, trying shorter amplicons in the order of 150-200 bp and play more with template concentrations to see if amplicon yield can be significantly improved. I suggest authors should be more emphatic to highlight limitations of their approach to pave the way forward more securely towards the implementation of necessary improvements.

Response

While we feel that many of the analytical challenges of this methodology have been addressed, we do acknowledge that some modification of the primer panel may be helpful in optimizing its use for certain viral types and that panel changes and/or optimization of the starting template may help in further improving genome coverage for some lineages (lines 647 to 651).

Minor observations

Line 234 I would suggest adding in parenthesis after “panel failed to support amplification” (due to differences in primer affinities across different variants and/or lineages).

Response

Addition made as suggested (see lines 281-282).

After lines 236-237 I think there could be a third hypothesis that may be considered, which would account for the availability of longer and higher concentrations of viral -SSRNA templates in unfixed tissues that in turn increases the probability for the yield of longer amplicons. Meanwhile, formalin fixed tissues may only have available short length -SSRNA templates, due to nucleic acid fragmentation during the fixation procedure, thereby yielding higher concentrations of shorter amplicons as observed in amplicons patterns on Figures 2 and 3. The species assignment analysis better supports this hypothesis over the one suggesting amplification of longer length could predominantly come from host DNA. Along the same lines poor sequence coverage in several samples may indicate less than ideal concentration of -SSRNA templates during the multiplex amplification, which need to be highlight by authors with more emphasis as an impediment to generate homogenous genome coverages over 90% in all samples tested.

Response

A sentence has been added (lines 320 to 322) to acknowledge the likelihood that RNA fragmentation precludes amplification of longer amplicons for FFPE samples.

Lines 244-245 authors indicate that samples with Ct values under20 yielded fainted amplicons profiles. This could be a clear indication that some samples may contain less than ideal quantities of -SSRNA compromising ideal whole genome coverage. Ideally, higher that 90%.

Response

In the discussion we examine the correlation of Ct values and final genome coverage. Although one could expect that starting with a high level of viral template should tend to result in a better outcome regarding genome coverage, there does not appear to be a clear relationship between the two. Other factors such as primer sequence or sample integrity may be playing some role in this. 

Line 311 I would suggest adding the word intergenic or non-coding region after G-L., “highly divergent G-L region” for “highly divergent G-L intergenic region” to make emphasis this is a no amino acid coding region.

Response

The word “intergenic” has been added to this sentence now on line 379.

Lines 364-373 present a remarkable finding. However, it is concerning that its placement in the overall tree topologies is not consistent throughout phylogenetic reconstructions and differs for that commonly accepted in the field. These indicates concerning analytical noises likely introduced for the use of less-than-ideal multiple sequence alignments, which could be due to the presence of radically different sequence gaps generated in the same sample tested during the guided assembling using different rabies reference sequences.

Response

Indeed, as made clear in our data (Table S5) use of different reference sequences for the guided assembling resulted in different coverages and thus differences in the gaps of the assembled alignments. The use of appropriate references for the guided assembly is critical for generating optimal aligned sequences that maximize genome coverage and yield robust phylogenetic grouping.

Lines 444-447 I think this sentence is somehow confusing since it is not quite consistent with the overall results. Although the implemented Ampliseq Illumina approach demonstrated an ideal viral amplicon enrichment (over 90%), that was not sufficient to attain ideal genomic coverages (an average of >50%), which compromise and confounded downstream lineage/variant inferences. I would current this statement accordingly.

Response

The line numbers here do not seem to correspond to the indicated text. However, the point raised has been revised in accord with the modified analysis.

Lines 454-456 I think this would have been more accurate is samples tested would have been analyzed separately rather than in blocks. Aligning samples with multiple gaps across their genomes imposes severe analytical challenges for phylogenetic reconstructions. Neighbor-joining reconstruction should only be used for gapless perfectly aligned multiple sequences alignments. Otherwise, more sophisticated reconstruction algorithms such as the Bayesian one implemented in the Beast package should be attempted.

Response

Once the assemblies were optimized NJ analysis did perform well with pairwise deletion of gaps across the alignment. Use of the more complex Bayesian algorithms may be beyond the capabilities of some investigators thus we decided to keep with the simpler option.

Lines 467-473 I think this paragraph should be rephrased to improve accuracy. Results presented throughout this investigation demonstrate that the diversity of RABV present in the Western Hemisphere requires the use of tailored sequence assembling by using reference sequences pertaining to the same lineage/variant expected, which complicates and limits the optimization and utilization of this approach as a high throughput technology, unless the analytical pipeline gets meticulously improved.

Response

The line numbers here do not seem to correspond to the indicated text. However, the improvement in the analytical pipeline has addressed this issue and the text has been revised accordingly.

Reviewer 2 Report

The authors described Illumina sequencing strategy for obtaining nearly complete rabies virus genomes from FFPE samples, and viral typing and epidemiological analysis of obtained sequence data.  The purpose of the research is clearly and comprehensively presented and justified.

By described approach, the authors managed to get more than 97% of viral reads and less than 2% host reads thus proving the method to be powerful. Furthermore, quite successful genome coverages were obtained to conduct epidemiological analysis. Based on obtained sequence data, the authors discovered the previously unknown circulation of the Mexican skunk variant in California.

The authors also correctly identified and presented the shortcomings:

-Limited coverage of G-L genome region is obtained

-The need to select appropriate reference sequences in the Ref Seq Assembly has been recognized

-Although a good approach, it currently only satisfies the exact typing of the variants circulating in the Americas. Therefore, generated Illumina primer panel will not be suitable for epidemiological analysis of clades circulating on other continents.

In conclusion, the described methodology (custom-adapted!) will find application for typing archival samples. The application of this methodology will certainly lead to new discoveries in the epidemiology of rabies viruses, so I consider this paper important in the field of rabies diagnostics and epidemiology, and suitable for publication.
